# Relative Density of SLM-Produced Aluminum Alloy Parts: Interpretation of Results

**Corinne Arvieu \*, Cassiopée Galy, Emilie Le Guen and Eric Lacoste**

Arts et Metiers Institute of Technology, University of Bordeaux, CNRS, Bordeaux INP, INRAE, I2M Bordeaux, F-33400 Talence, France; cassiopee.galy@u-bordeaux.fr (C.G.); emilie.le-guen@u-bordeaux.fr (E.L.G.); eric.lacoste@u-bordeaux.fr (E.L.)

\* Correspondence: Corinne.arvieu@u-bordeaux.fr; Tel.: +33-05-5684-5865

**Abstract:** Micrographic image analysis, tomography and the Archimedes method are commonly used to analyze the porosity of Selective Laser Melting (SLM)-produced parts and then to estimate the relative density. This article deals with the limitation of the relative density results to conclude on the quality of a part manufactured by additive manufacturing and focuses on the interpretation of the relative density result. To achieve this aim, two experimental methods are used: the image analysis method, which provides local information on the distribution of porosity, and the Archimedes method, which provides access to global information. To investigate this, two different grades of aluminum alloy, AlSi7Mg0.6 and AM205, were used in this study. The study concludes that an analysis of the metallographic images to calculate the relative density of the part depends on the areas chosen for the analysis. In addition, the results show that the Archimedes method has limitations, particularly related to the choice of reference materials for calculating relative density. It can be observed, for example, that, depending on the experimental conditions, the calculation can lead to relative densities higher than 100%, which is inconsistent. This article shows that it is essential that a result of relative density obtained from Archimedes measurements be supplemented by an indication of the reference density used.

**Keywords:** additive manufacturing; selective laser melting; porosity; relative density measurement; archimedes method; AlSi7Mg0.6 and AM205 alloys

---

## 1. Introduction

Metal Additive Manufacturing (AM) processes are now widely used as they complement other shaping processes (casting, machining, welding, etc.). They have many advantages, particularly because they can produce very complex shapes in their entirety and manufacture parts from materials with a gradient of properties or composition; they are able to incorporate several functions in the same part and the same manufacturing process; they do not require heavy equipment, and material wastage is limited. Metal AM fusion processes can be divided into those where the material is deposited directly and those on a powder bed where a laser (SLM) or an electron beam (EBM) selectively melts thin layers of metal powder (a few tens of micrometers thick), one after the other, after the first layer has been spread on a metallic substrate. Titanium, nickel, and aluminum alloys and also steels are the metals most commonly studied with SLM technology [1,2].

The main difficulties associated with SLM-produced aluminum alloy parts are related to their surface state and the presence of porosities in the final material [3]. There are many phenomena that can account for the presence of porosities. In particular, they may be due to (i) the entrapment of surrounding gases or evaporation of some of the alloying elements in the melt pools [4]; (ii) the presence of inclusions such as oxides [5]; (iii) particle spatter ejections in the melt pools [6]; (iv) variations in

thickness of the layers around the melt pools [7]; or (v) the operating parameters selected for the process [8]. In the literature dealing with SLM, image analysis (microscopy and tomography) and the Archimedes method are the main techniques used to estimate the porosity rates of metal parts produced by SLM; some recent studies have covered 316 L steel [9,10], Ti6Al4V [11] and AlSi10Mg [12].

The objective of our study is to discuss the limitation of the relative density results to conclude on the quality of a part manufactured by additive manufacturing. There are indeed limitations, especially associated with the choice of reference materials to interpret the results of relative density. To investigate this, we carried out an experimental study on two different grades of aluminum alloy: AlSi7Mg0.6 and AM205. At first, before we focus on the Archimedes method, our analyses confirm the literature results of the complementarity of methods used to analyze porosity rates by highlighting the difficulties in qualifying the material health of SLM-produced parts using these methods.

## 2. Materials and Methods

### 2.1. Materials

The two alloys used in this study are AlSi7Mg0.6 and AM205. The powder obtained by inert gas atomization consists of spherical particles (Figure 1) ranging from 25 μm (D10) to 60 μm (D90) in diameter for the 2 types of powder (Figure 2). The composition of the powders and the AlSi7Mg0.6 reference alloy are given in Table 1.

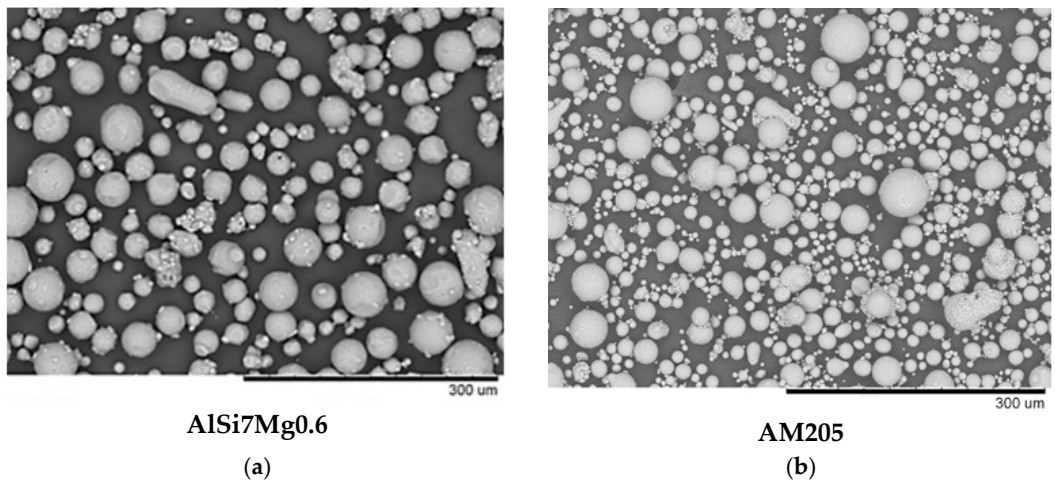

**AlSi7Mg0.6**

(**a**)

**AM205**

(**b**)

**Figure 1.** Particles of AlSi7Mg0.6 (**a**) and AM205 (**b**) obtained by gas atomization (SEM).

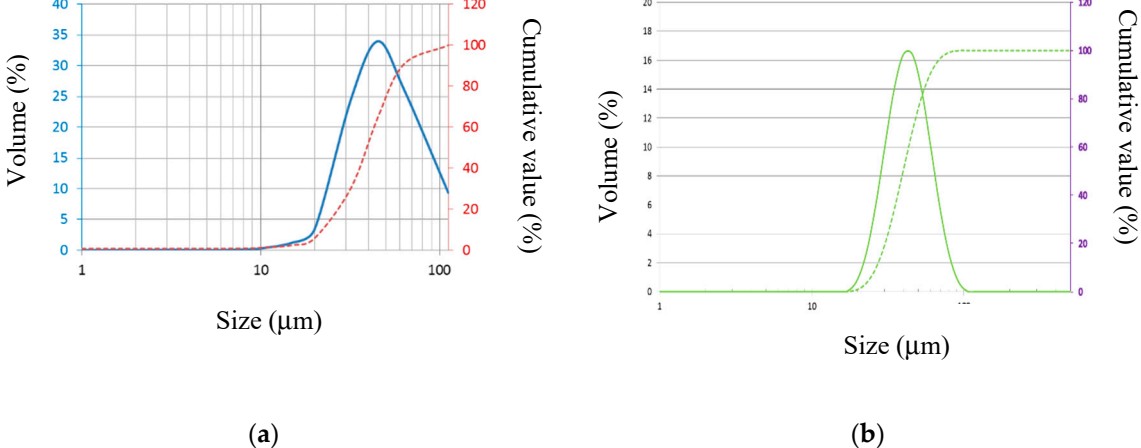

(**a**)

(**b**)

**Figure 2.** Size distribution of (**a**) AlSi7Mg0.6 particles and (**b**) AM205 particles.

**Table 1.** Chemical composition of the studied materials.

| Chemical Composition | Al | Si | Mg | Fe | Cu | Mn | Zn | Ti | Ag | B | Other |
|---|---|---|---|---|---|---|---|---|---|---|---|
| AlSi7Mg0.6 foundry alloy (Wt %) [ASM, 1993] | remainder | 6.5–7.5 | 0.21 | 0.15 | 0.03 | 0.1 | 0.07 | 0.10–0.18 | | | 0.1 |
| AlSi7Mg0.6 powder (Wt %) | remainder | 7.2 | 0.65 | 0.14 | <0.01 | <0.005 | <0.002 | <0.005 | | | 0.1 |
| AM205 powder (Wt %) | remainder | <0.10 | 0.20–0.33 | <0.08 | 4.2–5.0 | | | | 3.0–3.85 | 0.6–0.9 | 1.25–1.55 <0.17 |

## 2.2. SLM Process

The test specimens were produced by Selective Laser Melting on an SLM 280HL system (SLM Solution). This machine is equipped with an IPG Yb:fiber laser, wavelength $\lambda$ = 1070 nm, nominal power of 400 W and a focal diameter of 80 µm. The raw material is stored in a reservoir located above the level of the build zone. The recoater is gravity fed, with the amount of material needed to deposit two layers on the build zone. The AlSi7Mg0.6 alloy powder is then spread across the plate by a coating mechanism to form a thin, uniform layer on the build surface, with the plate preheated to 150 °C. The parts are built up under an argon atmosphere, mainly to avoid an oxidation phenomenon occurring during laser melting. The amount of oxygen present in the build chamber is less than 100 ppm.

All the test samples used in the study (AlSi7Mg0.6 alloy and AM205 alloy) were produced from new powder, using the same technique: first, for each layer, all the contours of the parts on the plate were lasered, then it was the turn of the interior of the parts using a stripe configuration. Irrespective of the material, the thickness of the layers, which is noted e, was 50 µm and the volume of all the final test samples was 15 mm × 15 mm × 15 mm.

The AlSi7Mg0.6 and AM205 test samples were produced using different sets of parameters obtained by varying the following parameters, where $e$ is the thickness of the powder layer: laser power $P$, laser speed $v$, and inter-layer distance h and therefore the volume energy density $\psi$ (Table 2), defined by the following Equation (1) [13]:

$$\psi = \frac{P}{e \cdot h \cdot v} \tag{1}$$

**Table 2.** Parameters used to produce AlSi7Mg0.6 and AM205 specimens.

| Alloy | Power W | Laser Spot Scan Rate mm·s$^{-1}$ | Scan Spacing mm | Energy Density J·mm$^{-3}$ |
|---|---|---|---|---|
| AlSi7Mg0.6 | 350 | 1150 | 0.272 | 22.38 |
| AlSi7Mg0.6 | 350 | 1150 | 0.221 | 27.54 |
| AlSi7Mg0.6 | 350 | 1150 | 0.145 | 41.98 |
| AlSi7Mg0.6 | 350 | 1150 | 0.119 | 51.15 |
| AlSi7Mg0.6 | 350 | 1150 | 0.068 | 89.51 |
| AM205 | 105 | 675 | 0.18 | 17.28 |
| AM205 | 400 | 1000 | 0.27 | 29.63 |
| AM205 | 225 | 625 | 0.18 | 40 |
| AM205 | 400 | 1000 | 0.09 | 88.89 |
| AM205 | 175 | 350 | 0.09 | 111.11 |

## 2.3. Micrographic Image Analysis

One method used to determine the relative density of a sample was image analysis. Analyzing micrographs is a destructive process used to characterize porosities both morphologically and

quantitatively. This method takes a polished surface and determines the ratio of the surface area that corresponds to porosities to the total surface area observed. For each polished section observed, the "real" image is transformed into a digital image via a camera. This digital image is made up of pixels containing specific information such as light intensity or color, which will then be converted into a grayscale. The information captured in each pixel is coded into a specific level of gray, with the user able to adjust the scaling. The next stage, thresholding, consists in transforming the digital image into a binary image. Depending on the gray level defined above, each pixel is converted into either black or white. The limit of separation is defined by the operator. By adjusting the threshold level, it is possible to extract the area specific to the porosities, within the resolution limit of the device. The ratio of this extracted area to the total area of the image gives the porosity rate of the sample, from which the relative density of the sample can be determined.

In our case, observations were made with a Keyence VHX 2000 series digital microscope equipped with a Z20U lens enabling bright-field observations with magnification ranging from 20 to 200. The depth resolution of the optical microscope (a few microns) requires a perfectly flat sample. Great care must be taken with the metallographic preparation before observation in order to obtain a flat, specular surface with no scratches, otherwise the count may be distorted.

The limit for observing small-sized porosities is directly related to the experimental conditions. We were therefore able to check that when using a magnification of X40, the minimum porosity surface that can be measured was around 25 $\mu m^2$, whereas it was 1 $\mu m^2$ for a magnification of 200.

## 2.4. Tomography

X-ray absorption tomography is a non-destructive control technique that reconstructs cross-sectional images of a three-dimensional object. With the progress in computers and optical equipment, a resolution of the order of one micron can now be achieved. The equipment used is a high-resolution Versa 500 Zeiss micro-imaging system, able to obtain images with pixels of 0.15 $\mu m$ and voltages of up to 160 kV. Tomography is a means of characterization that can reveal porosities within materials. It can locate porosities and describe their geometry. For example, this tool is used to characterize SLM-produced parts made of nickel alloy [14] or titanium alloy [11,15]. Maskery et al. used this tool to study the influence of heat treatments on the distribution and geometry of porosities, for SLM-produced parts made of Al–Si10–Mg [12].

## 2.5. Archimedes Method

The Archimedes method is used to measure the relative densities of parts developed by SLM [9]; it consists in weighing a single sample in two different fluids (Figure 3). Generally, the reference fluid is air. The second fluid is distilled water, acetone or ethanol. De Terris et al. showed that the choice of this second fluid had an influence on the density measurements of 316 L steel parts developed by SLM [10]. If distilled water is often used [16], it is not always suitable as air bubbles can form on the sample surface. This phenomenon is observed on lattice structures, for example, where air bubbles prevent water penetrating fully into the interior of the mesh because the surface tension of the water is too high [17]. Obaton et al., therefore, decided to use ethanol to measure the density of solid parts and lattice structures produced by SLM. In our study, we used ethanol 96%.

As the density of air is considered negligible, we determine that of the sample $\rho_{sample}$ according to an equation (Equation (2)) where $m_{air}$ is the mass of the sample weighed in air, $m_f$ is the mass of the sample immersed in ethanol and $\rho_f$ is the density of ethanol.

$$\rho_{sample} = \rho_f * \frac{m_{air}}{\left(m_{air} - m_f\right)} \tag{2}$$

Some authors have sought to modify this relationship by replacing, in the quantity $(m_{air} - m_f)$ of the expression (2), $m_{air}$ by the mass of the sample previously impregnated with liquid in the air, to take into account the open porosities in the density measurement [18].

Measurements were repeated three times for all the samples tested. No special sample preparation was necessary, apart from prior ultrasonic cleaning followed by drying, in order to eliminate any trace of residual powder.

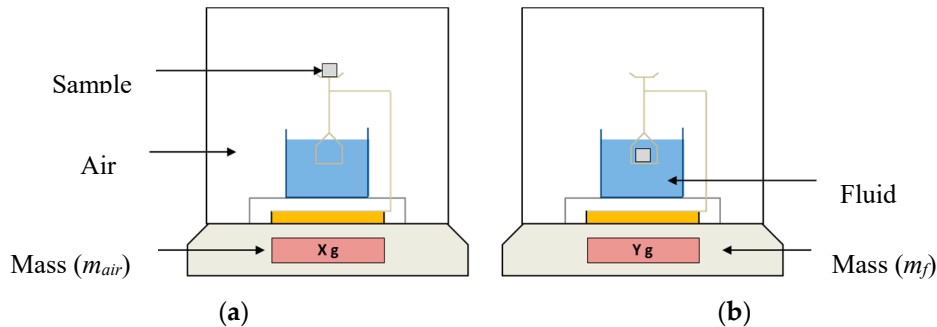

**Figure 3.** Principle of the Archimedes method: (**a**) the sample is first being weighed in the air (reference fluid) and (**b**) the sample is then being weighed in the second fluid.

### 2.6. Helium Pycnometer

True density is the density of the particles that make up a granular solid or a powder. If there is a significant difference between the true density of the powder and the theoretical density of the material of which it is constituted, this indicates the presence of defects within the powder, but gives no indication of the exact nature of the defects (intra-particle porosities, strong oxidation of the powder, defects in chemical composition, etc.).

True density is measured using a gas pycnometer Micromeritics AccuPyc 1340. This technique gives the real volume ($V_{true}$) of the sample. When the exact mass of the tested sample is known ($m_{sam}$), the true density ($\rho_{true}$) can be deduced from the following equation (Equation (3)):

$$\rho_{true} = \frac{m_{sam}}{V_{true}} \tag{3}$$

This technique uses the gas displacement method to measure volume precisely. A standard measure of gas, in our case helium, contained within a volume $V_E$ at pressure $P_1$ is placed in a cell of known volume $V_c$ containing the powder. The pressure measured is labelled $P_2$. The volume of the powder sample $V_p$ is then given by the following expression (Equation (4)):

$$V_p = V_c + V_E\left(1 - \frac{P_1}{P_2}\right) \tag{4}$$

## 3. Results and Discussion

### 3.1. Estimation of Relative Density by Image Analysis

The image analysis method prompts a few comments, suggesting that caution is required before attributing an intrinsic characteristic to a sample on the basis of this technique. Determining the porosity rate of a sample by image analysis is based on the assumption that what we observe on a 2D image is representative of the sample as a whole. Great care must therefore be taken with the sampling. This is essential in order to validate and generalize the results obtained to all of the part being analyzed.

Sections of AlSi7Mg0.6 samples produced by SLM display a random distribution of "small-sized" porosities (diameter ≤ 10 μm), while larger diameter porosities are distributed preferentially according to the laser strategy. As a result, the choice of cutting direction, the area selected, the preparation of

the sample, and the area considered for carrying out the count are all parameters that can affect the results obtained.

When measuring the relative density of our samples, we encountered a number of difficulties, mainly regarding the choice of magnification and the way in which the metallographic section of the test sample was produced.

### 3.1.1. Influence of Magnification

The choice of magnification for image acquisition is important on several levels. First, magnification determines the number of pixels in the image; the smaller the magnification, the smaller the number of pixels that make up the same surface area, as shown in the diagram in Figure 4. If magnification X40 produces a surface area S made up of a single image, for magnification X160, this will require 16 distinct images. The size of the pixel is very much reduced, thus making it possible to discern the smallest sized porosities, which could not be distinguished with a smaller magnification. In addition, the larger the magnification, the longer the acquisition and processing time for a surface area. Thus, the right compromise has to be reached to provide the desired accuracy in an acceptable time.

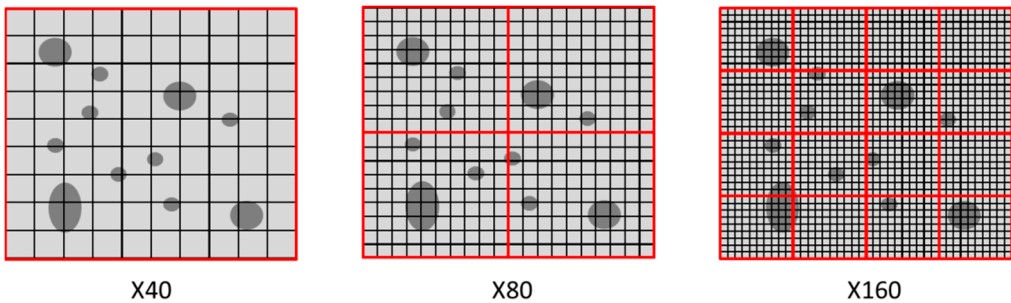

X40          X80          X160

**Figure 4.** Diagram showing the influence of magnification on pixel size.

To validate the procedure used here, we wanted to examine the influence of magnification on the levels of relative density obtained. To do this, images of the same polished surface of a section of the sample were analyzed at three different magnifications (X40, X80 and X200). Table 3 summarizes these results.

**Table 3.** Influence of magnification on relative density levels.

| Magnification | X 40 | X 80 | X 200 |
|---|---|---|---|
| Number of images needed to reconstruct the zone | 1 | 4 | 30 |
| Average relative density of the zone | 99.17% | 99.14% | 99.00% |

We observed that, in all three cases, the relative densities obtained were of the same order of magnitude, but tended to decrease as the magnification increased, which could be because the smaller porosities were being taken into account. However, it is important to note here that, in the case of the largest magnification, when scanning the same surface, an image reconstruction was necessary. This step was an additional potential source of error. Given the very small differences found in the same measurement of density, it is sensitive to conclude that magnification may have a significant influence. In this case, therefore, it does not seem necessary to use large magnifications. In the rest of the study, a magnification of X40 will be used.

### 3.1.2. Representativeness of the Surface Observed

Figure 5 shows microscopic sections taken in both directions (longitudinal and transverse) in relation to the build axis. We can see in the transverse direction relative to the build axis that there are areas where large porosities are concentrated, linked directly with the laser mode used. The choice of

surfaces analyzed on this section may therefore have an influence on the estimate for the porosity rate of the section as a whole.

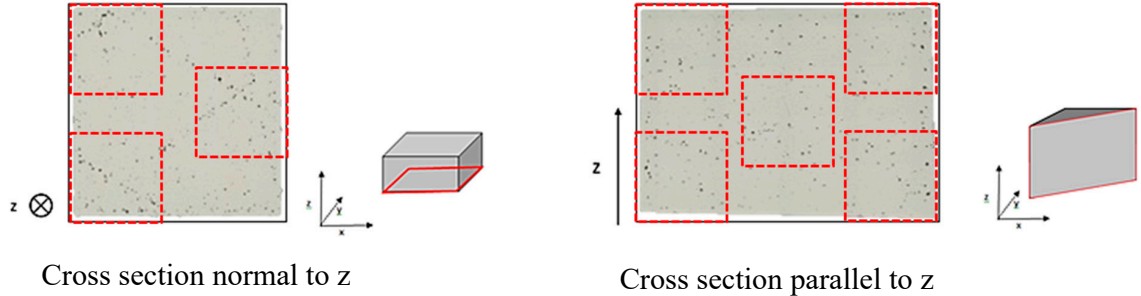

Cross section normal to z  Cross section parallel to z

**Figure 5.** Influence of cross section direction on sample representativeness (z is the build axis).

In this case, it is important to check whether measurement variabilities exist according to the zone studied. For example, if measurements are taken only in the area within the dotted lines in the cross-section normal to the building direction in Figure 5, with magnification X40, the average relative density from all the measurements from the selected zones is 98.52%. However, if the entire surface area is calculated at the same magnification, this gives a relative density of 98.33%. It should be noted here that analysis of the entire surface area would mean defining several partially overlapping zones, which would lead to further approximation.

In the case of the cross section parallel to the building direction, the distribution of the large porosities appears to be more homogeneous. Tests carried out on this section, which were similar to those for the cross section normal to the building direction, therefore showed a less pronounced variation: relative density varied from 98.60% for a calculation limited to the five zones defined in Figure 5 to 98.67% for measurements across the entire surface area.

In summary, as a result of this series of tests, it is difficult to conclude that the total surface area must be studied in order to estimate the porosity rate for a section. There is therefore a tendency to conclude that the choice of surface area has little influence on the result, provided that the surfaces chosen for observation are uniformly distributed across the entire surface area of the section.

However, determining the relative density of a sample on the basis of calculating the porosity rate by image analysis of a polished section assumes that the 2D image is representative of the volume of the sample. In addition to the direction of the section, it is necessary to check whether the density results are dependent on the positioning of the section studied. To do this, a series of analyses were carried out along build axis z in 0.5 mm steps obtained by successive polishing of the section along a length of 3 mm. The average relative density obtained from 3 images using the procedure previously described varied from 99.3 to 99.7% (Figure 6), which shows that the position of the section seems to be an influencing factor on the resulting relative density in agreement with the results of De Terris and all [10].

In order to analyze the differences observed, we considered the geometry of the porosities from tomographic observations with a low resolution of 2.4 µm. Figure 7 shows a 3D reconstruction of an observed parametric porosity. Clearly, the porosity is not spherical, and its walls are concave (on the right) and convex (on the left). It is therefore obvious that the orientation of the porosity with respect to the observed metallographic section will have an impact on the calculation of the area ratio.

It is clear that the ratio of the area extracted from the porosity to the total area of the image varies according to the section under consideration.

The non-spherical shape of the porosity therefore means that the results of an image analysis are dependent on the orientation of the porosity in relation to the section under consideration. This study is also able to focus on one of the limitations associated with analyzing metallographic sections of porosities: it is difficult to apply this method to determine density.

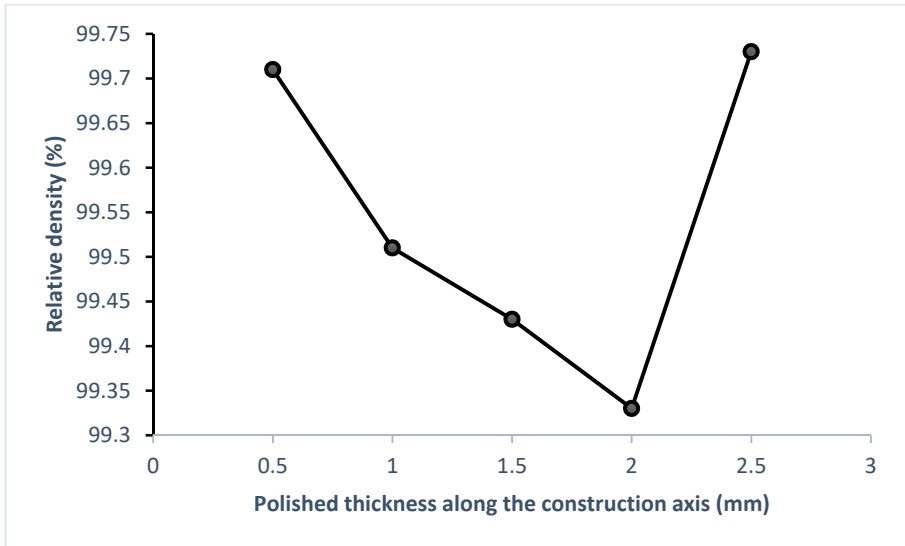

**Figure 6.** Example of change in relative density in an Selective Laser Melting (SLM) sample.

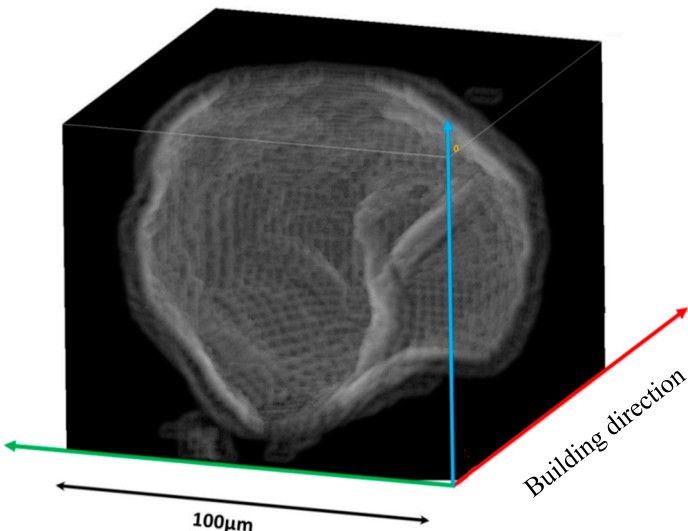

**Figure 7.** Three-dimensional reconstruction of a porosity.

Finally, we conclude that the analysis of metallographic images can produce information on porosity location, distribution and geometry. In addition, the observation of a section can also make it possible to detect the presence of powder that is not completely melted [10]. On the other side, tomographic analysis has the added advantage of providing a volume representation of the location and geometry of the porosities, but it has the disadvantage of being more complicated to implement and more costly.

However, with an analysis of metallographic images, calculating the relative density of the part depends on the areas chosen for analysis. This therefore limits the scope of this method for defining parts in terms of material health. As a result, to obtain volume and global information on SLM-produced parts, we used the Archimedes method.

### 3.2. Measuring Relative Density Using Archimedes Method

The relative density of sample $D_{relative}$ is given by the ratio of the density $\rho_{sample}$, obtained with the Archimedes method, to the theoretical density $\rho_{theoretical}$ of the reference.

This then raises the question of the choice of reference. In the literature, even in very recent studies, this choice is never discussed [10,18]. Yet this is an important question because the value of the theoretical density has a strong influence on relative density. In the case of the AlSi7Mg0.6 alloy, we can assume that the reference is the foundry alloy and hence that $\rho_{theoretical}$ = 2.68 [19]; however, we note that the composition of the foundry alloy and that of the powders are not perfectly identical (Table 1). We might then consider taking the powder material as the reference, as it has the same composition. In this last case, density $\rho_{theoretical}$ is the density of the powders, which is determined by helium pycnometry. However, the value obtained (2.66) is characteristic of the powder, while the particles of this powder have very clear porosities, as demonstrated by micrographic observation (Figure 8). The measured value is therefore not an intrinsic value of the material that makes up the powder and cannot therefore be selected as a reference.

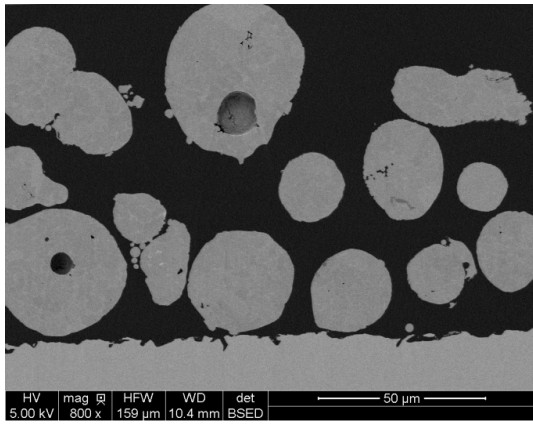

**Figure 8.** SEM observation of an AlSi7Mg0.6 powder cross section.

To conclude, in the case of the AlSi7Mg0.6 alloy, given that the differences observed in the chemical compositions are small, the traditional choice of a foundry alloy as a reference seems to be appropriate for assessing the material health of an AlSi7Mg0.6 part produced by SLM. The results obtained for the different sets of parameters, depending on the choice of $\rho_{theoretical}$, are given in Table 4. However, can this conclusion be applied to all grades of alloy?

**Table 4.** Change in relative density as a function of density for an AlSi7Mg0.6 part.

| Energy Density (J·mm$^{-3}$) | Relative Density with Foundry $\rho_{theoretical}$ | Relative Density with Powder $\rho_{theoretical}$ | Standard Deviation $\sigma$ (Foundry) |
|---|---|---|---|
| 22.38 | 95.53 | 96.25 | 0.2 |
| 27.54 | 97.31 | 98.05 | 0.17 |
| 42.12 | 97.98 | 98.71 | 0.008 |
| 51.15 | 98.10 | 98.84 | 0.004 |
| 89.51 | 98.20 | 98.94 | 0.03 |

The second aluminum alloy studied is AM205 (Table 1), an aluminum/copper/TiB2 alloy. The AM205 version has the "additive manufacturing" component, which resulted from the initial development of a high-resistance foundry alloy, A205. This alloy was developed with the aim of increasing the castability of alloy A201 while retaining the very high resistance properties of the original alloy. It can be considered as a metal matrix composite with a matrix composed of alloy A201, reinforced by the addition of titanium diboride TiB2 particles (between 2 and 4% by volume).

The true density of the AM205 powder that we measured by pycnometry was 2.88, which represents a considerable deviation from the density of the foundry alloy that corresponds to the grade studied,

i.e., that of the A205: 2.86 g.cm$^{-3}$. The relative density of the SLM samples was determined using the Archimedes method, according to the same protocol as for the AlSi7Mg0.6 alloy. As with the first alloy studied, we calculated this relative density in two ways: (i) by selecting the density of the A205 foundry alloy as the reference density, and (ii) by taking the density of the AM205 powders as the reference. Results on the change in the relative density of the samples as a function of the volume energy density $\psi$ applied when they were produced are given for the two reference densities in Table 5. It can be seen that, when the A205 foundry alloy is chosen as the reference, depending on the experimental conditions, the calculation leads to relative densities higher than 100%, which is inconsistent. This result once again highlights the importance of the question of the choice of the reference density for estimating the material health of an SLM-produced part: the two relative densities calculated on samples obtained by SLM do not have the same significance. In the case where the reference density is that of the casting alloy, the relative density can be used to compare two different manufacturing processes, casting and selective laser melting. However, with the powder as the reference, the results show the change in the material during the SLM process, as it changes from a powdery state to the continuous solid state.

**Table 5.** Change in relative density as a function of density for an AM205 part.

| Energy Density (J·mm$^{-3}$) | Relative Density with Foundry $\rho_{theoretical}$ | Relative Density with Powder $\rho_{theoretical}$ | Standard Deviation $\sigma$ (Foundry) |
|---|---|---|---|
| 17.09 | 87.72 | 88.33 | 0.25 |
| 29.63 | 98.53 | 99.21 | 0.006 |
| 40.00 | 99.26 | 99.95 | 0.007 |
| 88.89 | 99.58 | 100.27 | 0.014 |
| 111.11 | 99.70 | 100.40 | 0.008 |

It is therefore essential that a result of relative density obtained from Archimedes measurements be supplemented by an indication of the reference density used. Only then can comparisons with other results be validated.

## 4. Conclusions

SLM-produced parts are the subject of many studies to qualify and quantify their porosity rates and then to estimate the relative density. Although destructive image analysis methods provide local information on the distribution of porosity as well as the size of the porosity, this method is not suitable for calculating the relative density of the part because it depends on the areas selected for analysis.

The Archimedes method gives access to global information and therefore allows for estimating the relative density from the volume of the part. However, the Archimedes method has limitations because this method comes up against the problem of choosing the reference material used as a basis for calculating the relative density. In our study of two aluminum alloys, we have shown that when a material equivalent to the foundry alloy was chosen, despite it being traditionally used as a reference in the literature, this could result in relative densities greater than 100%.

It is then absolutely necessary when publishing the relative density obtained from the Archimedes method to specify the conditions for determining this value. It therefore seems important to mention that, irrespective of the method selected to estimate the material health of an SLM part in terms of porosities, this method nevertheless remains qualitative.

**Author Contributions:** Conceptualization, C.A. and C.G.; methodology, C.A. and C.G.; validation, C.A., C.G. and E.L.G.; formal analysis, C.A., C.G. and E.L.G.; investigation, C.A. and C.G.; data curation, C.A.; writing—original draft preparation, C.A. and E.L.; writing—review and editing, C.A. and E.L.; visualization, C.A. and E.L.; supervision, C.A. and E.L.; project administration, C.A., E.L.G. and E.L.; funding acquisition, C.A. and E.L. All authors have read and agreed to the published version of the manuscript.

**Funding:** This work is carried out within the framework of the FUTURPROD research project in collaboration with the region of Nouvelle-Aquitaine (France), ENSAM, ArianeGroup, Stelia Aerospace, PolyShape, and AGB.

**Conflicts of Interest:** The authors declare no conflict of interest.

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
