# Peer review of "Relative Density of SLM-Produced Aluminum Alloy Parts: Interpretation of Results"

_jmmp, doi:10.3390/jmmp4030083_

Round 1

Reviewer 1 Report

The aim of  this study was an among others of micrographic image analysis to analyze porosity of Selective Laser Melting produced parts. In the first sequence, the authors recalled and discussed about complementarity of the different methods and the difficulties in qualifying the material with these methods. And in the second sequence, they focused on the interpretation of the relative density result. The authours found that it is essential that a result of relative density obtained from Archimedes measurements be supplemented by an indication of the reference density used.

The presented research results of the authors are relatively interesting. The results of study are clearly and understandable described in manuscript and the number of citations is includes current publications. Sometimes it seems that the manuscript despite the study results has a review character, but this does not reduce its quality.

Before publishing should be consider the following comments:

  1. The abstract does not contain relevant information. The authors not specified clearly the purpose of the study, methods applied and main results of study obtained.
  2. The literature analysis is too general. Relevant data from selected publications should be descriptive in a few sentences.
  3. Axis descriptions in Figures 2a and 2b are cut off, and the figures in axle descriptions should be larger.
  4. The tables description should be above them.
  5. Formula 1 is well known. I propose to remove it.
  6. In some places in the Table 1 and in Figure 6 are commas instead of periods. This should be corrected.
  7. Figure 3 should be described. What does the reader see in the first and what in the second drawing?
  8. Descriptions in Figures 7 is cut off and it should be corrected.
  9. Formula 5 is unnecessary. Description in the manuscript text is sufficient.
  10. The conclusion chapter needs to be modified. Three important conclusions from the conducted research should be included.The summary is too general.
  11. The list of publications does not comply with the journal's requirements (JMMP tamplate). This should be corrected.

Author Response

Response to the reviewer 1

We have taken good note of the concerns expressed in the screening report on our first submission. We would like to thank the reviewer for your time spent for proof-reading our article.

Please find attachemnt our responses to the reviewer. We hope to satisfy all his requirements in the new version of the text.

Reviewer 2 Report

This paper makes the conclusion that it is very difficult to measure the density of a material prepared by selective laser melting to provide information on material health and porosity. The authors focus on micrographic image analysis and Archimedes method and demonstrate that both have issues. I have a couple of issues that need to be addressed.

The authors mention tomography but did not actually do any experiments with it from what I can tell. Move section 2.4 elsewhere as it was not done experimentally in the paper.

I would like to see a table in the conclusions with the pros and cons of the 3 methods: image analysis, tomography, and Archimedes and estimates of the error bars associated with each. Also include the ‘cost’ of the measurement. This would be a real benefit to the community. Overall, a nice piece of work that can be improved for greater impact. This work will be very useful for practical applications.

Author Response

Response to the reviewer 2

We have taken good note of the concerns expressed in the screening report on our first submission. We would like to thank the reviewer for your time spent for proof-reading our article.

Please find attachment our responses to the reviewer. We hope to satisfy all his requirements in the new version of the text.
